# Optimizing Reusable Knowledge for Continual Learning via Metalearning

**Julio Hurtado**
Department of Computer Science
Pontificia Universidad Católica de Chile
`jahurtado@uc.cl`

**Alain Raymond-Saez**
Department of Computer Science
Pontificia Universidad Católica de Chile
`afraymon@uc.cl`

**Alvaro Soto**
Department of Computer Science
Pontificia Universidad Católica de Chile
`asoto@ing.puc.cl`

## Abstract

When learning tasks over time, artificial neural networks suffer from a problem known as Catastrophic Forgetting (CF). This happens when the weights of a network are overwritten during the training of a new task causing forgetting of old information. To address this issue, we propose MetA Reusable Knowledge or MARK, a new method that fosters weight reusability instead of overwriting when learning a new task. Specifically, MARK keeps a set of shared weights among tasks. We envision these shared weights as a common Knowledge Base (KB) that is not only used to learn new tasks, but also enriched with new knowledge as the model learns new tasks. Key components behind MARK are two-fold. On the one hand, a metalearning approach provides the key mechanism to incrementally enrich the KB with new knowledge and to foster weight reusability among tasks. On the other hand, a set of trainable masks provides the key mechanism to selectively choose from the KB relevant weights to solve each task. By using MARK, we achieve state of the art results in several popular benchmarks, surpassing the best performing methods in terms of average accuracy by over 10% on the 20-Split-MiniImageNet dataset, while achieving almost zero forgetfulness using 55% of the number of parameters. Furthermore, an ablation study provides evidence that, indeed, MARK is learning reusable knowledge that is selectively used by each task.

## 1 Introduction

Among the cognitive abilities of humans, memory is one of the most relevant. However, memories are fragile. In particular, studies from cognitive psychology show that memories can be lost when new knowledge disrupts old information, in a process known as task interference [1, 2]. In the case of artificial neural networks (ANNs), task interference is behind the problem known as Catastrophic Forgetting (CF) [3, 4, 5, 6, 7]. In effect, as a model learns a new task, its weights are overwritten causing catastrophic forgetting of old information.

Unfortunately, in the case of ANNs, task interference is more severe than in the case of humans. This is evident for the case of continual learning, *i.e.*, the case when data from new tasks is presented sequentially to the learner, who does not have access to previous data. Under this training scheme, when a previously trained model is retrained on a new task, it usually suffers a significant drop in performance in the original task [8, 9, 10]. Previous works to prevent CF have followed two main strategies. The first strategy [10, 8, 11, 12, 13] consists of avoiding the modification of parameters

35th Conference on Neural Information Processing Systems (NeurIPS 2021).

that are key to solve previous tasks. Specifically, when facing a new task, a regularization term ensures that critical parameters are modified as little as possible. In general, this approach shows satisfactory performance in problems that involve few tasks, however, when the number of tasks increases, problems such as accumulated drifting in weight values and interference among them, make this approach difficult to scale. The second strategy [14, 15, 16, 17, 7, 18] consists of introducing structural changes to the architecture of a model. This includes methods that reserve part of the network capacity to learn each task [9], and methods that use special memory units to recall critical training examples from previous tasks [17, 19]. The main problems with these methods are the extra model complexity and the need of an efficient method to recall key information from previous tasks.

In contrast to these previous strategies, when learning new tasks, humans continuously associate previous experience to new situations, strengthening previous memories which helps to mitigate the CF problem [20]. Taking inspiration from this mechanism, we propose MetA Reusable Knowledge or MARK, a new model based on a learning strategy that, instead of mitigating weight overwriting or learning independent weights for different tasks, uses a metalearning approach to foster weight reusability among tasks. In particular, we envision these shared weights as a common Knowledge Base (KB) that is not only used to learn new tasks, but also enriched with new knowledge as the model learns new tasks. In this sense, the KB behind MARK is **not given by an external memory that encodes information in its vectors**, but by a *trainable model that encodes shared information in its weights*. As a complementary mechanism to query this KB, MARK also includes a set of trainable masks that are in charge of implementing a selective addressing scheme to query the KB.

Consequently, to build and query its shared KB, MARK exploits two complementary learning strategies. On the one hand, a metalearning technique provides the key mechanism to meet two goals: i) encourage weight updates that are useful for multiple tasks, and ii) enrich the KB with new knowledge as the model learns new tasks. On the other hand, a set of trainable masks provides the key mechanism to selectively choose from the KB relevant weights to solve each task.

In terms of its inner operation, MARK works by first forcing the model to reuse current knowledge via functions that detect the importance of each pattern learned in the KB, and later expanding its knowledge if past knowledge is not sufficient to successfully perform a task. As an intuition behind how MARK works (not to be taken literally), when facing different tasks related to the classification of animals, MARK may take advantage of its metalearning approach to learn relevant features to discriminative concepts such as fur, paws, fin, and ears. Afterwards, when solving a task related to the discrimination of horses vs zebras, it might use a mask to focus on features related to fur rather than fins or ears.

We perform extensive experiments to analyze the inner workings behind MARK. In particular, we demonstrate that, indeed, MARK is able to accumulate knowledge incrementally as it learns new tasks. Furthermore, in terms of the pieces of knowledge used to solve each task, our experiments confirm that the use of task dependent feature masks results critical to the success of the method. We summarize our contributions as follows:

- We propose MARK, a novel method for continual learning scenarios that is based on two complementary mechanisms: i) A new method to build a KB that incrementally accumulates relevant knowledge from several tasks while avoiding CF problems, ii) A mechanism to query this KB to extract relevant knowledge to solve new tasks.

- An extensive experimental evaluation and public implementation of MARK that demonstrate its advantages with respect to previous works.

## 2   Continual Learning Scenario

Following previous work on continual learning [21], we consider a task incremental scenario, where task identity is provided at training time. Each task $t$ consists of a new data distribution $D^t = (X^t, Y^t, T^t)$, where $X^t$ denotes the input instances, $Y^t$ denotes the instance labels, and $T^t$ is a task ID. This task ID is required during training and inference. The goal is to train a classification model $f : X \longrightarrow Y$ using data from a sequence of $T$ tasks: $D = \{D^1, ..., D^T\}$. Following the usual setup for continual learning [21], each task is presented sequentially to the model. Our scenario allows for unlimited use of data from the current task, but after switching to a new task, this data is no longer available. Furthermore, there is no intersection between the classes of different tasks, but they share a similar domain. This domain intersection is key to take advantage of common patterns among

the tasks. Next section presents the details behind MARK, our proposed model for incremental learning in the previous scenario.

## 3 Method

When learning tasks sequentially, humans build upon previous knowledge, leading to incremental learning. In contrast, in a similar scenario, ANNs devote all of their resources to the current task, leading to the problem of CF. Taking inspiration from the behavior of humans, an appealing idea is to provide the model with a KB that, as the model faces new tasks, incrementally captures relevant knowledge. Using this shared KB, the model can associate previous experience to new situations, mitigating the CF problem. To implement this idea, we have to address two main challenges: i) How do we build this KB incrementally?, and ii) How do we query this KB to access relevant pieces of knowledge?

To address the first challenge, we leverage metalearning in order to train a KB from data. Specifically, we use a metalearning strategy known as episodic training [22]. This strategy consists of sampling from a task pairs of support and query sets to simulate training data from a large number of mini-tasks. These mini-tasks are then used to bias the learner to improve its ability to generalize to future tasks. In turn, this generalization leads to our goal: to capture relevant knowledge that can be reused to face new tasks.

To address the second challenge, on top of the KB, we train mask-generating functions for each task. These masks provide a suitable mechanism to selectively access the information encoded in the weights of the KB. We can envision these masks as a query that is used to access the intermediate activations of the KB. Following [23], these masks depend solely on each specific task and input. In this way, given an input, MARK uses the corresponding mask to query the KB generating a feature vector. This vector is then used by a task dependent classification head to output a prediction. We describe next the details behind this operation.

### 3.1 Model Architecture

Figure 1 shows a schematic view of the operation behind MARK, where the main modules are:

- **Feature Extractor** ($F^t$): This module is in charge of providing an initial embedding for each input $X_i$, i.e., $F^t$ takes input $X_i$ and outputs a vector representation $F_i^t$. In our case, we test our method using visual recognition applications, therefore, $F^t$ is given by a convolutional model. It is important to note that model $F^t$ can be shared among tasks or it can be specific to each task.

- **Knowledge Base** ($KB$)**:** This is the main module behind MARK. It is in charge of accumulating relevant knowledge as the model faces new tasks. In our implementation, we use a convolutional architecture with $B$ blocks. This part of the model is shared across tasks.

- **Mask-Generating functions** ($M^t$)**:** These modules take as an input feature vector $F_i^t$ and produce an instance and task-dependent mask $M_i^t$ for each block of the KB. Each mask consists of a set of scalars, one for each channel in the convolutional blocks of the KB, that multiply the activation of each channel. These masks are critical to select what knowledge is relevant to each instance and task. In our implementation, we use fully connected layers.

- **Classifier** ($C^t$)**:** These modules correspond to task-dependent classification heads. Its input $F_{i,KB}^t$ is given by a flattening operation over the output of the last block of the KB. Given the task ID of an input $X_i$, the corresponding head outputs the model prediction. In our implementation, we use fully connected layers.

Specific details can be found in Section A.1. As shown in Figure 1, the flow of information in MARK is as follows. Input $X_i$ goes into $F^t$ to extract the representation $F_i^t$. This representation is then used by $M^t$ to produce the set of masks that condition each of the blocks in the KB. The same input $X_i$ enters the mask-conditioned KB leading to vector $F_{i,KB}^t$ used by the classification head. Finally, classifier $C^t$ generates the model prediction, where $t$ is the task ID associated to input $X_i$.

### 3.2 MARK Training

Algorithm 1 describes the training process behind MARK. The first step consists of initializing the KB by training it end-to-end on the first task, without using metalearning and mask functions. In

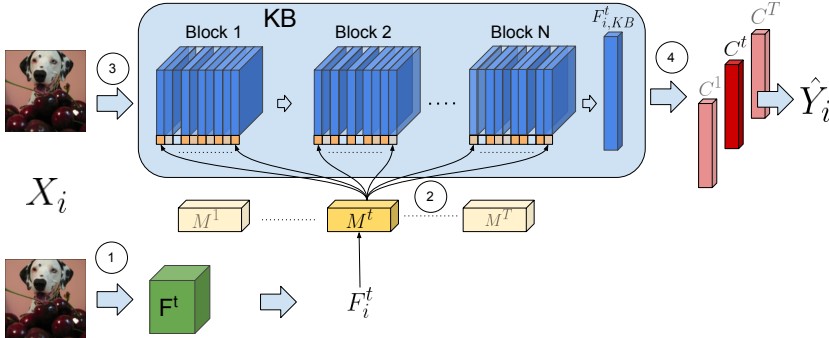

Figure 1: Given an input $X_i$ from task $t$, (1) We use a feature extractor $F^t$ to obtain $F_i^t$. (2) $F_i^t$ is then passed to mask function $M^t$ to generate mask $M_i^t$. Afterwards, (3) the same input $X_i$ enters the KB, which has intermediate activations modulated by $M_i^t$. Finally, (4) the modulated features go through a task-dependent classifier $C^t$ that performs the class prediction for $X_i$.

other words, we perform the KB initialization using the regular training procedure of a convolutional neural network for a classification task. After this, we train MARK sequentially on each task by alternating three main steps:

1. **KB Querying.** We train task-dependent mask-generating functions that are used to query the KB using vector $F_i^t$. Also, we concurrently train the task classifier for the current task. Notice that, leaving aside the KB initialization, in this step each new task is trained using only accumulated knowledge from previous tasks.

2. **KB Update**. We use a metalearning strategy to update the weights in the KB. This scheme allows fostering KB updates that favor the acquisition of knowledge that can be reused to face new tasks.

3. **KB Querying.** After updating the KB using knowledge from the current task, we repeat the querying process to finetune the mask-generating functions and task classifier using these new knowledge. Notice that during this step the KB is kept fixed.

The intuition behind the application of the three previous steps is as follows. We initially query the KB using the accumulated knowledge from previous tasks. This forces mask functions and classifiers to reuse the available knowledge. When that knowledge is exhausted, we proceed to add knowledge from the current task into the KB. Finally, we take advantage of this newly updated KB to obtain our final mask-functions and classifiers for a given task. We describe next the main steps behind the process to query and update the KB.

---

**Algorithm 1:** MARK - Training Process

**Components:**
- $D^t$: Dataset for task $t$.
- $F^t$: Feature extractor for task $t$.
- $KB$: Knowledge Base.
- $M^t$: Mask-generating function for task $t$.
- $C^t$: Classifier for task $t$.

$KB \leftarrow Train\{KB(D^1)\}$    /*Init KB */
**for** *task* $t \leftarrow 1$ **to** $T$ **do**
     /*1. Obtain feature vectors $F_i^t$*/
     $F_i^t \leftarrow \{F^t(D_i^t)\}, i \in \{1, \ldots, |D^t|\}$
     /*2. Train $M^t$ and $C^t$ using current KB*/
     $Train\{M^t \ \& \ C^t\}$
     /*3. Update KB using metalearning*/
     $KB \leftarrow$ KB-Update    /*Algorithm 2*/
     /*4. Finetune $M^t$ and $C^t$ with updated KB*/
     $Train\{M^t \ \& \ C^t\}$
**end**
**Output:** Trained modules $KB, M^t, C^t$.

---

**Algorithm 2:** KB-Update

**Components:**
- $KB$: Knowledge Base.
- $C^k$: Temporal Classifier for batch $k$
- $E_{outer}$: number of training updates for KB (outer loop)
- $E_{inner}$: number of inner loop epochs.

**for** $e$ *in* $E_{outer}$ **do**
     Generate K batches of data from $D^t$
     **for** $k$ *in* $K$ **do**
         $KB^k \leftarrow KB$ /*Copy of $KB$*/
         $Initialize(C^k)$
         Train $KB^k$ and $C^k$ with batch $k$ for
         $E_{inner}$ epochs.
     **end**
     $\nabla KB \leftarrow \frac{1}{E_{inner}} \sum_k^K \gamma_k (KB^k - KB)$
     $KB \leftarrow KB - \alpha \nabla KB$
**end**
**Output:** $KB$    /*output updated $KB$*/

### 3.2.1 KB Querying

Once we obtain feature vectors through the use of feature extractor $F^t$, the model can learn which are the modules in the KB that can best solve the current task. In this training stage, the model trains functions to learn how to use the knowledge available in the KB, focusing solely on reusing knowledge from previous tasks, *without modifying the KB*. In particular, in this step we only train $M^t$ and $C^t$. Both are trained end-to-end, while keeping the KB weights frozen.

As we generate masks for each intermediate activation of our model, strictly speaking we have a total of $B$ mask-generating functions. However, to facilitate the notation, we subsume all such functions under the term $M^t$ and think of its output as the concatenation of the results of those $B$ functions. Eq. 1 shows the function $M^t$, where a mask $M_i^t$ is obtained given an input $X_i$ from task $t$. Our implementation of these functions consists of a linear function with parameters $W^{t,M}$, and an activation function $\rho$, which we implement as a ReLU.

$$M_i^t = M^t(F_i^t) = \rho((W^{t,M})^T F_i^t) \tag{1}$$

Masks generated in this process are encouraged to have two effects: first, to give a signal of how important a specific module from the KB is for the current input; second, to make sure that gradient updates are done where they really matter. If an activation map is irrelevant for a certain task, then the value of the corresponding mask will be zero, making the gradient update associated to that activation being zero as well.

### 3.2.2 KB Update

The purpose of this training step is to add new knowledge *from the current task* to the KB. As a further goal, we aim to enrich the KB with features that are highly general, i.e., they can be used to solve several tasks. To achieve this, we use metalearning as a way to force the model to capture knowledge that can be reused to face new tasks.

Figure 2 shows an schematic view of the metalearning procedure that we use to train MARK. This procedure consists of an adaptation of the training process presented in [24]. Specifically, we create a set of $K$ mini-tasks, where each mini-task consists of randomly sampling from the current task a set of $H$ classes and $h$ training instances per class. This allows us to create a mini-task that differs from the main task, finding weights not specific to it. We train one copy of the model using each mini-task for $E_{inner}$ epochs. We refer to the copy $k$ trained for $e$ epochs as $KB_e^k$. For each mini-task we use a temporary classifier $C^k$ initialized with the parameters of $C^t$. We discard this classifier after the final iteration of the inner-loop.

Following [25], our method for episodic training consists of two nested loops, an inner and an outer loop. The inner loop is in charge of training the copies of our KB for the current mini-task while the outer loop is in charge of updating the KB weights following a gradient direction that leads to fast adaptation to new mini-tasks. During each inner loop, $KB^k$ and $C^k$ are trained end-to-end for $E_{inner}$ epochs.

To simulate the outer-loop and update the $KB$, we follow Eq. 2. Specifically, for each $k$, we average the difference between the parameters of the $KB$ before $KB_0^k$ and after $KB_{E_{inner}}^k$ in the inner loop (see Eq. 2). This is simply an average of the sum of the cumulative gradients for each model $KB^k$. This is repeated $E_{outer}$ times.

$$KB = KB - \alpha \nabla KB \qquad \nabla KB = \frac{1}{E_{inner}} \sum_k^K \gamma_k (KB_{E_{inner}}^k - KB_0^k) \tag{2}$$

The subtraction in Eq. 2 is weighted by $\gamma_k$. We compute $\gamma_k$ as shown in Eq. 3, taking as a reference the accuracy of each model on a validation batch from the same task $t$.

$$\gamma_k = \frac{acc_k}{\sum_j^K acc_j} \tag{3}$$

By updating weights with a weighted average of solutions of all meta-tasks, we expect that the new values of the full model should be (on average) an appropriate solution for different tasks.

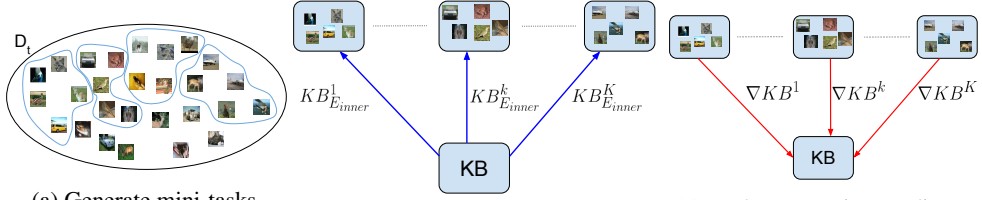

(a) Generate mini-tasks.    (b) Train copy of KB in each mini-task.    (c) Update KB using gradients.

Figure 2: KB update using metalearning. a) Given a task $t$, we randomly generate a set of $K$ mini-tasks, where each mini-task consists of a subset of classes from the original task. b) For each mini-task, we train an independent copy of the current KB for a fixed amount of epochs, leading to $K$ models. c) Afterwards, we calculate gradients with respect to the loss function of each of these models using a hold-out set of training examples. Finally, we update the KB using a weighted average of these gradients.

## 4 Experiments

For our experiments, we use two benchmarks used in previous works [18, 8, 26]. The first one is 20-Split CIFAR-100 [27] that consists of splitting CIFAR-100 into 20 tasks, each one with 5 classes. The second one is 20-Split MiniImagenet [22] that consists of dividing its 100 classes into 20 tasks.

In our experiments, we compare MARK with recent state of the art methods: Hard Attention to the Tasks (HAT) [7], A-GEM [28], Adversarial Continual Learning (ACL) [18], GPM [29], Experience Replay and SupSup[30]. We also include a Multitask baseline to serve as an upper bound for performance, where all tasks are trained jointly. In the Appendix, we include a comparison of the size of these alternative models with respect to MARK. All hyperparameters used for each method are also included in the Section A.2.

In terms of the feature embedding $F^t$ used to represent input instances, we test the following approaches:

- **MARK-Task:** we train $F^t$ for each task adding a classifier on top of it that is trained using $D^t$. After training $F^t$, this classifier is discarded.
- **MARK-Random:** $F^t$ consists of a set of random weights. All tasks share the same $F^t$.
- **MARK-Resnet:** all tasks share a Resnet-18 pre-trained on Imagenet as a feature extractor.

We run all of our experiments using 3 different seeds. In terms of hyperparameters, we use SGD with a learning rate of 0.01 and a batch size of 128. Each task is trained for 50 epochs. To update the KB, we use 10 meta-tasks ($K$), trained for $E_{inner}$ =40 epochs, each with a learning rate of 0.001. We repeat this training stage 20 times ($E_{outer}$). Code is released at: `https://github.com/JuliousHurtado/meta-training-setup`.

### 4.1 Metrics

To quantify the performance of our method, we use two metrics: average accuracy (Acc) and *backward transfer* (BWT). These are given by:

$$Acc = \frac{1}{T} \sum_{i=1}^{T} Acc_{T,i} \qquad BWT = \frac{1}{T-1} \sum_{i=1}^{T-1} Acc_{T,i} - Acc_{i,i} \qquad (4)$$

Acc measures average performance over the T tasks after the sequential learning. BWT measures how much performance is lost on previous tasks after sequential learning. As a measure of efficiency, we also consider the amount of memory used by each method, considering number of parameters and extra temporal information needed by each method.

### 4.2 Results

We first analyze the general performance of our method with respect to the alternative approaches. We highlight the version of MARK where we pre-train $F^t$ using data from each task (MARK-Task).

Table 1: Results using 20-split CIFAR-100 and 20-split MiniImagenet datasets. Standard deviation for 3 runs is listed in parentheses.

| Method | CIFAR100 | | | Mini-Imagenet | | |
|---|---|---|---|---|---|---|
| | Acc (std.)% | BWT% | Mem% | Acc (std.)% | BWT% | Mem% |
| HAT | 76.96 ($\pm$ 1.2) | **0.01** | 146% | 59.45 ($\pm$0.1) | -0.04 | 198% |
| A-GEM | 61.88 ($\pm$ 0.2) | -16.97 | 205% | 52.43 ($\pm$3.1) | -15.23 | 165% |
| ACL | 78.08 ($\pm$ 1.3) | 0 | 134% | 62.07 ($\pm$0.5) | 0 | 195% |
| GPM | 76.67 ($\pm$ 3.1) | -0.42 | 66% | 60.41 ($\pm$0.6) | 0 | 70% |
| Exp. Replay | 65.90 ($\pm$ 1.2) | -16.9 | 130% | - | - | - |
| SupSup | 77.58 ($\pm$ 1.3) | 0 | 26/111%[1] | - | - | - |
| Multitask | 84.08 ($\pm$ 1.2) | 0 | 159% | 74.44 ($\pm$1.6) | 0 | 99% |
| MARK-Random | 60.91 ($\pm$ 0.4) | -1.73 | 41% | 33.02 ($\pm$0.7) | **1.29** | 43% |
| MARK-Task | **78.31** ($\pm$ 0.3) | -0.27 | 100% | **69.43** ($\pm$1.6) | -0.39 | 100% |
| MARK-Resnet | 86.29 ($\pm$ 0.1) | -3.05 | 345% | 93.55 ($\pm$0.2) | -2.52 | 126% |

While Table 2 shows that MARK-Resnet leads to better results, it uses additional information due to its pre-training using ImageNet. It is important to note that embeddings $F_i^t$ are not directly used to classify the corresponding instances. Instead, they are applied to generate the masks used to query the KB. Aside from its impact on accuracy, we observe no meaningful impact of the input representation on BWT.

As shown in Table 2, for both datasets, MARK-Task outperforms all competing methods in terms of average accuracy, while showing no sign of CF (BWT is close to zero). This is especially remarkable for Mini-Imagenet, as competing methods use complex AlexNet-based architectures during training versus our simple convolutional model. It is also noteworthy that MARK-Task accomplishes this while using almost half less memory than its closest competitor, ACL. This is because MARK-Task reuses the same stored KB for all tasks, while for each task it needs to store a small mask-generating function and classifier. Other methods either need to store extra parameters for adversarial training [18] or they require access to past gradients [28][29].

### 4.2.1 Evolution of Weight Updates

We hypothesize that the positive results achieved by MARK are due to the knowledge stored in its KB being highly reusable. To test this, we analyze the updated weight during training of all tasks. As a baseline, we consider a version of MARK that does not include metalearning and mask-filtering steps. In this experiment, weights are considered updated if their average deviation is over a certain threshold after training a task. Please see the appendix for specific details.

Figure 3 shows the percentage of weights that change after training each task. We observe two distinct phenomena: 1) MARK changes a rather small number of weights compared to the baseline, and 2) As more tasks are trained, the amount of weight updates dwindles.

We attribute both phenomena to MARK's ability to reuse previous knowledge to tackle new tasks. Thus updates should be needed mostly when learning new knowledge. The dwindling amount of updates can be linked to MARK's ability to develop its KB incrementally, thus new tasks are far more likely to be tackled with knowledge already in the KB. We believe that task similarity plays a crucial part in this behaviour, with each new task learned facilitating MARK's ability to find similarities for future tasks. However, to elucidate how important is task similarity to achieve these results is beyond the scope of this study. We leave this as an interesting avenue for future work.

### 4.2.2 Task Learning Speed

We hypothesize that if the KB is storing reusable knowledge, then learning speed for new tasks should increase as the KB incrementally contains more knowledge. We analyze the accuracy curves for each task when using MARK versus a model that is sequentially trained without including any mechanism

---

[1]Memory usage of SupSup depends on whether the model is generated from a seed or stored fully.

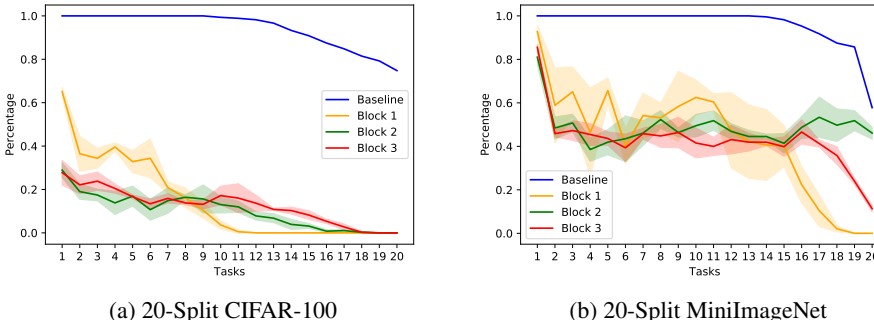

(a) 20-Split CIFAR-100       (b) 20-Split MiniImageNet

Figure 3: Percentage of updated weights in different blocks of the KB during sequential training. As a baseline, we consider a version of MARK that do not include metalearning and mask-filtering. Using MARK fewer updates are required, suggesting that new tasks add less knowledge to the KB due to incremental learning.

to avoid CF. Given that MARK trains its masks and classifiers two times per task, to be fair, we train the baseline for twice as many epochs as MARK. Consequently, we report results for the baseline every two epochs. Figure 4a shows the comparison in speed between the two models. We observe that indeed, on average, MARK achieves higher accuracy and stabilizes quicker than the baseline. This provides further evidence about the ability of MARK to encode reusable knowledge.

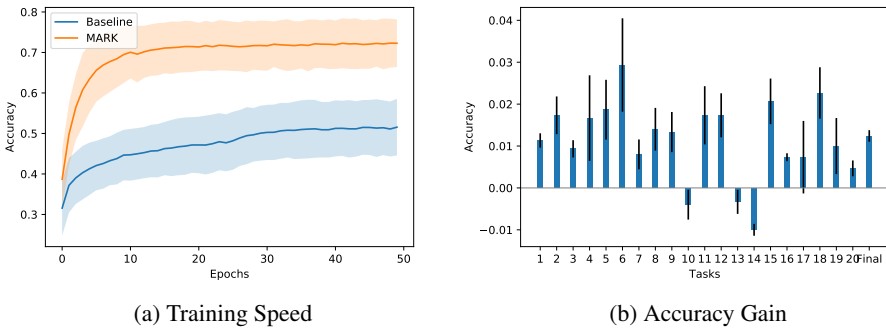

(a) Training Speed       (b) Accuracy Gain

Figure 4: (a) Average test accuracy in the CIFAR-100 dataset achieved by MARK and a baseline model that is sequentially trained without including any mechanism to avoid CF. MARK achieves both greater accuracy and faster stabilization. (b) Accuracy in a task when we re-training the task after the model complete its sequential learning using all available tasks. We observe an increase in performance for most tasks, suggesting successful knowledge accumulation in the KB.

### 4.2.3 Importance of Metalearning and Mask Functions

We study the impact of the metalearning strategy used to update the KB and the mask-generating functions used to query the KB. To do this, we compare MARK against three ablations:

- **Baseline:** Simple sequential learning with no metalearning or mask-generating functions. We use the same architecture as the KB.
- **Baseline + ML:** We improve the baseline by adding metalearning, i.e., KB update.
- **Baseline + Mask:** We improve the baseline by adding task-specific mask functions.

Figure 5 shows that the baseline suffers from both significant forgetting and reduced performance. In contrast, when we include metalearning (*Baseline + ML*) forgetting is reduced. Similarly, when we only add mask-functions (*Baseline + Mask*), there is a boost in performance but forgetting also increases. By using metalearning and mask-functions, our full MARK-Task model achieves high accuracy with almost no forgetting.

Two important observations can be extracted from these experiments: 1) The forgetfulness of *Baseline + ML* is more significant than that obtained in MARK ($-3\%$ v/s $-0.25\%$), showing that learning to use prior knowledge through masks can also help reduce forgetfulness. 2) The maximum accuracy averaged over tasks (without forgetting) is higher in MARK than in *Baseline + Mask* ($78\%$ v/s $74\%$), showing that training with metalearning helps knowledge transfer between tasks. This behaviour might seem counterintuitive as we metalearn only using the current task. However, an important part of MARK is forcing the reuse of previously useful features which biases solutions to start from a point which is useful for previous tasks. It is also worthwhile to note that it is known [24] that a REPTILE-style update (as is used in MARK) optimizes for synergistic gradient updates between minitasks. This might also help learn features that are more reusable, which might explain part of the success of MARK.

### 4.2.4 Incremental Construction of KB

We expect that, as MARK faces new tasks, its KB should be enriched with new knowledge. To test this hypothesis, we analyze the effect of training again each task after the model completes its sequential learning using all available tasks. As expected, Figure 4b shows that indeed, for most tasks, there is an absolute increase of 1.24% in performance when they are revisited after completing the sequential learning cycle. However, this increase might have to do with relearning what was forgotten rather than learning new knowledge. Thus, we also compared the difference between the maximum accuracy achieved during training for a task versus its final accuracy after retraining. We observed an absolute 0.97% average increase in accuracy as well, which shows that indeed new knowledge was used by earlier tasks.

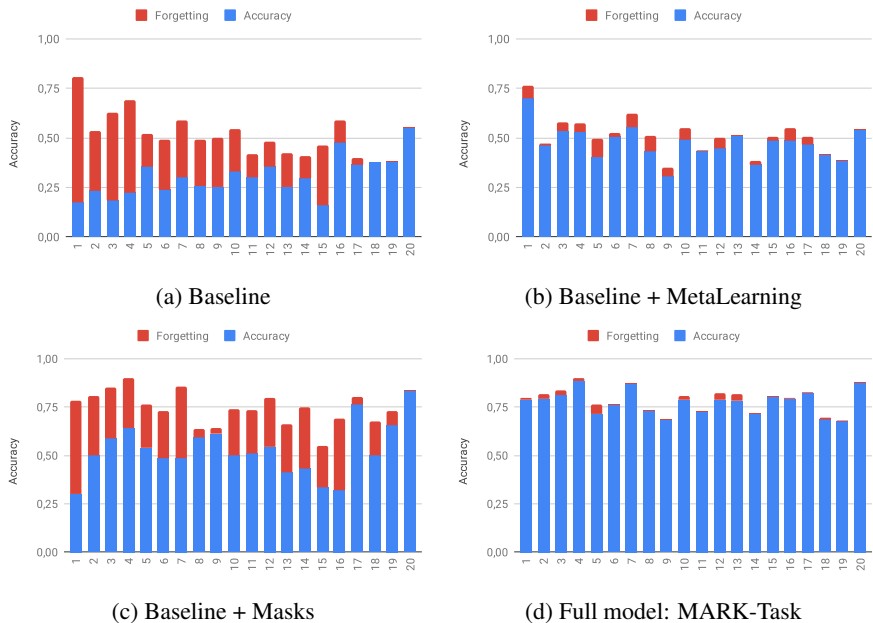

(a) Baseline          (b) Baseline + MetaLearning

(c) Baseline + Masks          (d) Full model: MARK-Task

Figure 5: CF and accuracy for different versions of MARK-Task tested on 20-Split CIFAR100. (a) Without using metalearning and mask-functions, performance is low and CF is high. (b) Adding only metalearning, performance is still low, but there is almost no forgetting. (c) Adding only mask-functions, performance increases but forgetting is still high. (d) MARK-Task, our full model, achieves high performance with almost no forgetting.

## 5    Limitations

Our method relies on a base assumption that there is common structure to be learned between tasks. Thus, if this does not happen it will perform poorly (as any method which expects to reuse parameters for different tasks). Therefore, MARK is sensitive to the quality and structure of the first task.

As our method relies on task-specific modules, the amount of parameters scales linearly with the amount of tasks, possibly hindering its practical use in settings with a large amount of tasks.

It is not a straightforward task to extend our model to use memories for replay, since our focus has been on the scenario where we can't have access to past data after switching tasks. It is unclear where replay should happen, nor what should be stored as both masks and samples could work well.

# 6 Related Work

Previous works have tackled the problem of CF using different strategies. Memory-based methods mitigate CF inserting data from past tasks into the training process of new tasks, continuously retraining previous tasks [31], either with raw samples [32, 26], or minimizing gradient interference [33, 28]. Other works such as [34, 35] train generator functions (GANs) or autoencoders [36] to generate elements from past distributions. They seek memory-efficiency by generating examples. Similarly, [37, 38] seek to be memory-efficient by saving feature vectors of instances from previous tasks, while learning a transformation from the feature space of past tasks to current ones.

Other methods focus on limiting the flexibility of new tasks to modify the current model. The typical approach penalizes weight modifications or freezes a subset of the model. This can be achieved by adding weight regularizations [10, 8] or using masks to freeze parts of the model [16, 15]. Another alternative is to increase network capacity by adding extra parameters [9, 18]. Works similar to ours have proposed to find a path of relevant weights to solve the task [39], freezing used weights, and limiting learning of new tasks. Others use different functions as components in the network, either Hypernetworks [40], Deep Artificial Neurons (DANs) [41], or Compositional Structures [42], so that the network components are more flexible during learning.

Metalearning [43] is the idea of learning to learn. It has been used as a training strategy in sequence learning [44, 45] and continual learning scenarios [46, 47, 48, 49, 50]. In the case of [44, 45], the objective of meta-learning is to learn a starting point from where future sequences can learn sub-components of the model in the so-called inner-loop. In the case of MARK, we use the meta-learning strategy while learning new tasks, encouraging the reuse of the weights before and after the modification of the KB.

The idea of conditioning the weights of a model has being used before. [51] normalizes the output of a convolution layer by squeezing the activation maps via linear functions. [23] conditions the output of convolutional layers using the question in a VQA problem. However, as far as we know, this idea has not been combined with metalearning to update a KB.

# 7 Conclusions

In this work, we present MARK, a novel method for continual learning scenarios that is based on the construction and query of a KB that uses metalearning to incrementally accumulate relevant knowledge from different tasks. Our experiments indicate that the use of metalearning to build the KB is the crucial factor to mitigate CF, while the use of mask-functions to query the KB is the key factor to achieve high performance. Specifically, MARK achieves state of the art results in both 20-Split CIFAR-100 and 20-Split MiniImageNet, while suffering almost no BWT. Compared to ACL, the best performing alternative method, MARK uses 55% of the parameters, 51% of the memory requirements, while achieving almost a 12% increase in average accuracy on the 20-Split MiniImageNet benchmark.

## Acknowledgments and Disclosure of Funding

This work was supported by National Agency of Research and Development (ANID) via National Doctorate Scholarships. This project has also received funding from the Millennium Institute of Foundational Research on Data (IMFD).

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
