# OpenReview forum: "Optimizing Reusable Knowledge for Continual Learning via Metalearning"
_NeurIPS.cc/2021/Conference — NeurIPS 2021 Poster_

### Official Review · Reviewer_ZFES · 2021-07-14

**Rating:** 6
**Confidence:** 4

**Summary:**

This paper develops a method (MARK) for tackling catastrophic forgetting in neural networks that maintains a knowledge base (KB) in the form of a convnet, which is shared across training of all tasks, and which, when combined with dynamic feature masking and a form of meta-learning to update the KB, enables MARK to outperform competing methods when trained on sequential image classification tasks. In more detail, MARK classifies an image as follows: the image is passed through a task-specific feature extractor, producing an embedding which is used to generate masks (via a task-specific MLP) over the feature maps of the KB convnet (through which the image is passed separately), the output of which is then passed to a task-specific classifier. When a new task is encountered, the first step (“KB querying”) updates the task-specific parameters over several epochs, while keeping the KB parameters fixed; the second step updates the KB parameters using the meta-learning algorithm Reptile, where the inner loop updates the KB parameters and the task-specific classifiers over a batch of “mini-tasks”, each of which are created by sampling a subset of images from a subset of classes in the current task. Finally the KB querying task is repeated before moving to the next task. Experiments are conducted on 20-way split cifar100 and split-mini-imagenet, showing that MARK outperforms a number of competing methods in terms of average accuracy and forgetting, while using less memory in total. Experiments are run using different types of feature extractor (random fixed and one pretrained on imagenet), and with ablations of the meta-learning update and the mask-generating features, demonstrating that the former is most important for mitigating forgetting, and the latter for achieving a good maximum task performance.


**Limitations And Societal Impact:**

There is not much discussion of the limitations of the paper. It is worth mentioning that, while the method uses less memory than the methods compared to, it does suffer from the fact that the number of parameters increases with each task, which may not be scalable for a large number of tasks, especially since the task-specific feature extractors are not insignificantly small.
There is a section in the appendix detailing the CO2 impact of the experiments.


**Main Review:**

The paper introduces a novel method that combines meta-learning and dynamic feature masking to yield strong empirical results in a couple of standard continual learning benchmarks, and thus it would be a useful contribution to the CL community. For this reason, I recommend it for acceptance, but only marginally, predominantly because several design choices and details of the implementation are not clearly motivated, and I was left unsure about why the model works, despite the effort made in the paper to demonstrate the importance of each component with ablations. Below are more detailed comments:
- If one were following a typical meta-learning setup, one would expect the KB querying step to match the inner loop of the meta-learning KB update step, but there are at least two differences: (i) the KB parameters are kept fixed in the KB querying step, but are updated in the inner loop of the KB update, and (ii) the mask-generating features are updated in the KB-querying step but not in the inner loop of the KB update step. Could the authors explain the rationale behind these design choices that seem to differ, as far as I am aware, from the usual meta-learning setup (particularly (i))?
- In the ablation experiments, it is curious to see that, while the mask generating functions have a large impact on maximum performance per task, meta-learning the KB is most important for reducing forgetting. What is the intuition here? One would intuitively expect meta-learning to impact the speed of learning of new tasks (as is shown in the paper), but why should it reduce forgetting?
- The KB update
    - Why are the updates weighted by gamma_k? Does this not encourage more adaptation to tasks that the model is already good at?
    - What if after the first KB Querying update, the model is already good at performing task t - is there a danger of it not getting a signal at all in the KB update?
- In the baselines that ablate the meta-learning update, is the KB update still updated using SGD or is it kept fixed after training on the first task? If it is kept fixed, then an important ablation is missing, which is to train the KB with SGD along with the rest of the parameters; otherwise, we don’t know how important the meta-learning is for the performance of the model.
- The KB is trained fully on first task. What if first task is not that relevant for the subsequent ones? Maybe an ablation with a randomly initialised KB could be useful to understand the importance of this?
- In Figure 4b, is the accuracy gain over the maximum performance achieved in that task, or over the final performance having trained on all tasks? If it’s the latter, to what extent can you attribute this gain to just recovering forgotten knowledge, as opposed to new knowledge accumulated over training of other tasks?
- Algorithm 2 is a bit confusing:
    - S_F is not initialised.
    - C_k is not mentioned here, but in the main text it says that C_t is copied and trained as well in the inner loop.
- Even though it makes use of extra data, it is encouraging to see the MARK-Resnet (where the feature encoder is pretrained on Imagenet) perform so well in a sequential setting.
- Comments on related work:
    - It is mentioned in line 293 that previously “metalearning has not been used to increment a KB as in our model”. However, in [1] and [2], meta-learning is used to update a representation learning network or a neuromodulatory network, both of which can be thought of as a knowledge bases. In [1] the representation learning network is used directly in inference, and in [2] the neuromodulatory network gates the representation learning network, which bears similarities to the dynamic masking in MARK.
    - As well as HAT, which is cited in the paper, two other methods that use masks for continual learning are [3] and [4], which might be worth mentioning.

Minor comments:
- Line 199. The name of the referred method is wrong here: “Hard attention tasks” -> “Hard attention to the task”
- Line 217 and 230. “BTW” -> “BWT”
- Line 230 “specially” -> ”especially”
- Line 264. “both,” -> “both”
- Figure 3 y axes seem to be mislabelled - the range is from 0 to 1 but the axis labelled as “percentage” - should it not be “proportion”, or otherwise the scale should be from 0 to 100?

[1] Javed, Khurram, and Martha White. "Meta-learning representations for continual learning." Proceedings of the 33rd International Conference on Neural Information Processing Systems. 2019.

[2] Beaulieu, Shawn, et al. "Learning to Continually Learn." ECAI 2020. IOS Press, 2020. 992-1001.

[3] Wortsman, Mitchell, et al. "Supermasks in Superposition." Advances in Neural Information Processing Systems 33 (2020).

[4] Wen, Yeming, Dustin Tran, and Jimmy Ba. "BatchEnsemble: an Alternative Approach to Efficient Ensemble and Lifelong Learning." International Conference on Learning Representations. 2019.


**Time Spent Reviewing:**

8

---

> ### Author Response · Authors · 2021-08-08
> **R4**
>
> We appreciate the comments received, below we answer each of them:
>
> **Main Review**:
>
> 1. Thank you for your comments! There seems to be a misunderstanding on the relation of the KB-Querying step to our metalearning solution: it happens before and after metalearning, not during metalearning (KB-Update). Therefore, mask finetuning is not part of either the inner loop nor outer loop. Having said that, we will discuss your concerns in more detail:
>     1. During the inner loop of the KB update step, the KB parameters are not updated. What is updated are the *copies of the KB* where each is assigned to a different minitask. Later, in the outer loop of the KB update step, the KB is finally updated based on a weighted average of the metagradients derived from each of the copies.
>     2. The idea behind not updating the mask functions during the inner loop is twofold: first, it incentivizes solutions that can still work well for previous tasks and second, more practically, adjusting knowledge (KB) and where to look for it (Masks) at the same time is very slow and unstable. We believe that the three stages of sequential training, KB-Querying - KB-Update - KB-Querying, help to support the synergistic combination between the metalearning and mask training steps.
>
> 2. We understand the concern of the reviewer, the metalearning step (KB-update) is applied only to data from the current task, therefore, it is surprising its relevant effect to avoid CF, as shown in Figure 5. We believe that this is due to two main reasons: i) As shown in previous works, the use of an episodic training strategy fosters weight reusability. In other words, training based on an inner and outer loops that operate over mini-tasks, biases the learner to improve its ability to generalize to further tasks, fostering the learning of more generic rather than task-specific features. This impacts not only future but also previous tasks; ii) During metalearning, the feature selector masks are kept fixed, therefore, only the KB features that are relevant to solve the current task are updated. We believe that this is a key mechanism to mitigate CF because gradient updates are performed only where they really matter, helping to preserve previous knowledge. If an activation map is irrelevant for a certain task, then the value of the corresponding mask will be zero, making the gradient update associated to that activation being zero as well.
>
> 3. 1. Good observation. As we only run one model per minitask we are unable to distinguish between a bad model or a hard task. Thus, we chose to bias our solution to models that work well on their given tasks. There is an underlying assumption that we expect minitasks to be of roughly the same difficulty. We did try to use different weighting mechanisms: in particular we tried using uniform weighting and using the opposite of gamma, i.e., (1 - gamma). When using uniform weights, we find that the solution achieves 77.78% BWT: -0.37% . By using (1-gamma) we find the solution achieves 77.8% BWT: -0.09%. So we find that weighting is slightly better using our current method.
>     2. This could certainly happen, but it would mean there is no extra information from task t that the model hasn’t already learned from previous tasks. This would be the best performing case for our model, as there would be 100% knowledge reuse.
>
> 4. In that specific ablation when we don’t use metalearning, the KB is being updated using SGD. The main objective of this ablation is precisely to compare and verify how metalearning behaves in our architecture.
>
> 5. If by a randomly initialized KB you mean using a random KB for the first task instead of training it fully on the first task, then yes, we have tried that. We found that this slows the training process. In particular, the first tasks attempt to re-use random knowledge from the KB which affect the learning process.
>
> 6. This is a very good observation. This is measured over the final performance having trained on all tasks (accuracy at the end of iteration 1 versus accuracy at the end of iteration 3), thus we are unable to distinguish using this figure between which type of knowledge is being acquired. If we carry out a more complete analysis as suggested by the reviewer, we can observe two important details:
>     1. By comparing the performance at the end of training task 20, for the first time. We can observe that the BWT obtained by the model (-0.27%) is much smaller than the gain in accuracy (1.24%), showing that even if we recover all that was forgotten, the gain is greater.
>     2. If we compare the best performance of each task in the first iteration. We can observe that on average the performance increases by 0.97%.
>
> 7. 1. This is a typo. S_F = KB.
>     2. You are correct, we will fix the algorithm to include the creation of the C_k based on C_t.
>
>
> **Comments on related work**:
>
> 1. These works, OML and ANML, correspond to references [43] and [44] in our paper. There are important differences between these methods with respect to the inner working of our work, however, we agree that it is possible to consider that they are also incrementally updating a KB, especially in the case of OML. We will modify our paper accordingly and we will also include a description of the main differences with respect to these works.
>
> 2. We agree and will update our Related Work with these. We have added [3] as a baseline as well.
>
> **Minor comments**:
>
> We appreciate your attention to these details. We will fix this.
>
> **Limitations And Societal Impact:**
>
> We agree with you and will add a Limitations section on the papers with this and other limitations we have found.

---

### Official Review · Reviewer_fktr · 2021-07-15

**Rating:** 5
**Confidence:** 5

**Summary:**

The paper proposes a new architecture for task-based continual learning. It comprises two main parts :
1) a Knowledge Base (equivalent to a regular CNN) and
2) a Mask function with task-specific weights, which generates a per-example set of masks to apply on the weights of the knowledge base.
With the masked weights and the task-specific head a prediction is then obtained.

To train the model (for tasks > 1), the authors propose to alternate between training mask function keeping the KB fixed, then updating the KB with a fixed mask function. For the KB update, the authors develop a Reptile-like algorithm where artificial "tasks" are created by performing multiple updates on data coming from a subset of the classes in the current tasks.

The authors evaluate their method in the multihead setting on split CIFAR100 and MiniImagenet datasets. They further provide two ablations to evaluate how their contribution affects training speed and how many weights are changed.

**Ethical Concerns:**

No ethical concerns

**Limitations And Societal Impact:**

Yes / ok

**Main Review:**

__Originality__:  This works presents a novel combination of well known techniques (masking for CL, reptile-like updates for CL). It is my opinion that some similar work should be cited for this submission

[1] Wortsman, Mitchell, et al. "Supermasks in superposition." NeurIPS 2020. The authors learn a set of task-specific binary masks over a fixed set of weights.  The authors also look at similar datasets, I strongly recommend that the authors add this baseline to their paper.

__Quality__:  The method presented is sound. __My main criticism of the paper lies in the experimental section__. For all (3) baselines considered, the authors took numbers from the respective papers. This is not problematic in itself, _provided all papers use the exact same training and evaluation protocol_. This is not case the case here, as for example AGEM works in the single epoch setting (resulting in ~2% of the compute used for the author's 50 epoch setting). Moreover, no information is given as to how hyperparameter selection was performed (was a similar compute budget given to each method ?) Overall the empirical section does not meet the bar for this conference.

For me to accept this paper, the authors should
-> Include more baselines (at the very least [1]). Expansion based methods (e.g. Progressive Neural networks) also perform very well in the setting explored by the authors, and they have no forgetting by design. A simple replay based method should also be included.
->  Be very explicit as how the hyperparameter selection for different methods was performed.


__Clarity__: Overall the paper is well written and easy to follow. The figures are well made as well.

__Significance__:  The setting explored by the authors is very relevant (even if only two datasets are explored)


**edit** : after having clarified some of my concerns, I have increased my score by one point


**Time Spent Reviewing:**

2.5h

---

> ### Author Response · Authors · 2021-08-08
> **R3**
>
> We appreciate the valuable comments, we expect our response will help to alleviate your concerns regarding our experimental section.
> Regarding training and evaluation protocols: The results shown for previous methods are on equal terms with ours, either in number of epochs, number of parameters, type of model, and optimizer. We will expand section A.2 on the appendix to include information about the hyperparameters used for each method. For now, this is the list of hyperparameters used:
> * MARK: We use no scheduler, SGD, learning rate: 0.01, Momentum: 0.9, Weight Decay: 0.01. Batch size: 128. These settings are the same as for ACL.
> For the KB querying step: we train for 50 epochs, twice. For the KB update step: Inner Loop: 40 epochs, Outer Loop: 15 epochs. We set momentum to 0 in this step.
> * ACL: we take results directly from the paper. They train for 200 epochs. Learning rate: 0.01, Batch Size: 128, SGD.
> * HAT: we take results directly from the ACL paper. We assume hyperparameter choice between HAT and ACL to be fair as results are taken from the same paper. Therefore, it should be fair with MARK.
> * AGEM:  For CIFAR100, we use an implementation from Avalanche [1]. We use a memory of 200 per task and run it for 100 epochs. For MiniImageNet we use the results from the ACL paper. This paper mentions that their AGEM implementation was modified to be multi-epoch. They used 200 epochs and 13 images per class for the memory. Learning rate: 0.01, SGD, Batch Size: 128.
> * GPM: We slightly modified their code to run CIFAR100 with 20 tasks. We ran it for 100 epochs using their own code. Learning rate: 0.01, Batch Size: 128, SGD. Our KB model is very similar to theirs so we don’t replace it. We obtained Acc: 76.67% (std 3.1%) and BWT: -0.42%. For MiniImageNet, we take results directly from the paper. We were unable to find hyperparameters used either on the paper or code base.
> * Experience Replay: We run it for 100 epochs. We use a memory of 250 elements per task. We use 250 to make sure that the memory requirements between ER and MARK are similar. Batch Size: 128, SGD, Learning Rate: 0.01. CIFAR100: 65.9% (std 1.24%) and -16.9% BWT.
> * SupSup [2]: We used their code. Then, we tried different values of sparsity until we were able to replicate the same results of their paper with their setup. Using that sparsity value, we replaced their model with a model similar to our KB but with extra parameters, similar to the model used in ACL and GPM, and then we ran it for 100 epochs for 3 different seeds. Results are slightly below MARK: CIFAR100:  77.58% (1.3% std) and 0 BWT.
>
> **Extra baselines**: Thank you so much for referring us to these works. We have added SupSup and Experience Replay to our baselines. We did not include Progressive Neural Networks as we believe ACL is  a similar method while being closer to the state of the art. Due to the short time since the delivery of the reviews, we only managed to run experiments using  CIFAR100. These are the results:
> Experience Replay: CIFAR100 65.9% (std 1.24%) BWT: -16.9%, SupSup:  CIFAR100 77.58% (std 1.3%) BWT: 0
>
>
> [1] Lomonaco, Vincenzo, et al. "Avalanche: an End-to-End Library for Continual Learning." Proceedings of the IEEE/CVF Conference on Computer Vision and Pattern Recognition. 2021.
>
> [2] Wortsman, Mitchell, et al. "Supermasks in superposition." NeurIPS 2020

---

> > ### Comment · Reviewer_fktr · 2021-08-10
> > **Re**
> >
> > Thank you for providing clarifications on your experimental procedure. I appreciate the author's commitment to answering my comments and running additional baselines.
> >
> > I want to reiterate that my concern is not regarding what the actual hyperparameters are, but rather that each baseline was given a similar hyperparameter search budget. I understand that most of the results were taken from the ACL paper, which is ok. I encourage the authors to provide more information in the main paper on the provenance of the results.
> >
> > Please change the following sentence in the appendix  **"numbers for HAT, A-GEM and ACL are taken from the respective articles"**. This suggests that the AGEM results were taken from AGEM and not from ACL.
> >
> > Finally, I am not sure I understand the reasoning regarding the sparsity value for SupSup. A reasonable thing to do would be to first fix the architecture with which you plan on running the method **and then** do a hyperparameter search.
> >
> > Lastly, I still believe the empirical section is not as strong as it could be. Again, it's hard to compare different methods across different architectures, especially given how close the accuracy numbers are for CIFAR100.
> >
> > I have increased my overall score by 1 point given these clarifications.

---

### Official Review · Reviewer_FTM8 · 2021-07-16

**Rating:** 6
**Confidence:** 5

**Summary:**

The paper presents a continual learning method that uses a shared network (knowledge base) to learn combined knowledge across multiple tasks. This shared network is trained using meta-learning, a version similar to Reptile, on the dataset of the current task. In addition to the knowledge base, the paper proposed parameterized masking operators for each task, that mask out the irrelevant features in the knowledge base for a given task. The overall training proceeds in three steps. In step 1, the knowledge base is kept fixed, and the masking network and the classifier for the given task is trained. In step 2, for the trained masking network and the classifier, the knowledge base is updated using a meta-learning approach. Finally, in step 3, the masks and classifiers are finetuned using the updated knowledge base. Experiments are reported on Split-CIFAR 100 and miniImageNet dataset, where the proposed method is shown to perform better than the reported baselines. Some ablation of the method is also performed where it is shown that both the mate-learning and masking contribute to the overall better performance of the algorithm.


**Main Review:**

Positives:

- The paper is written well and easy to understand.
- The proposed approach strongly outperforms the baseline.
- Most of the ablations are done nicely and gives a better picture of why the proposed method is working.

Negatives/ Questions:

- Motivation for meta-learning: The motivation of using meta-learning is not clear to me. The way it has been used in this work, meta-learning is only performed at a given task-level (by splitting the task into tiny subtasks). In my opinion, the assumption that meta-learning would help different tasks exploit the common structure would only work if the training involves all those tasks jointly. As per my understanding of the paper, the authors perform meta-learning incrementally. The reason they still see the strong results could be that, in the benchmarks they considered, different tasks are not really different (different cifar-100 classes etc.). In a setup, where tasks differ the proposed method might see the forgetting problem in their knowledge base.

- Too many Task-specific parts in the approach : Currently the authors assume a task-specific feature extractor, mask and the classifier head. I wonder whether the method would even need a knowledge base. Perhaps two kinds of ablations can be performed here; 1) combine the feature extractor, masking network and the classifier for each task, train it end-to-end and report numbers, 2) add the knowledge-base, as the authors currently have, but keep it random (i.e) don’t train it, meta- or otherwise, and see how the network performs. It is difficult to discern from section 4.2.3 whether the reported ablations do any of the above. I see that the appendix makes an effort to understand a similar phenomenon but both the ‘feature’ and ‘no-retraining’ baselines differ slightly to what I am describing here.

- Architecture details: Please add the forward pointer to the appendix for the architecture details of all the components (knowledge-base, mask generator, feature extractor etc).

- Section 2:  Section is sparse in the details of the setup. Please add information regarding whether the authors assume an online access to the data of each task (single-pass setup). At test time what are the constraints, do the authors assume an identifier for the corresponding task.

- Baseline comparisons: Some of the methods, AGEM for example, might not be directly comparable as it assumes a single-pass through the data.

- Why baseline + MetaLearning would decrease forgetting: Do the authors have any intuition on why just the vanilla meta-learning on top of baseline would reduce forgetting, especially the way it is done (in a task-specific manner, see my comments above)? I would expect masking to be a better inhibitor of forgetting.


**Time Spent Reviewing:**

3

---

> ### Author Response · Authors · 2021-08-08
> **R2**
>
> We appreciate the comments received:
>
> Main Review:
>
> 1.  - There are 2 main reasons for our decision to only use minitasks derived from the current task: i) Our problem setting establishes that we do not have access to data from previous tasks, as mentioned in Section 2;  ii) Before the application of the metalearning step (KB-Update), during the KB-Querying step our method forces the model to reuse knowledge from previous tasks to solve the current task. Thus, our metalearning solution does not start from a blank slate, but rather from a solution that reuses previously relevant features. Furthermore, during the metalearning step, **the feature selector masks are kept fixed, therefore, only the KB features that are relevant to solve the current task are updated. If an activation map is irrelevant for a certain task, then the value of the corresponding mask will be zero, making the gradient update associated to that activation being zero as well**.  As a result, as shown in Figure 5, metalearning plays a critical role to avoid CF. Furthermore, as shown in Figure 4(a), it also helps to add incremental knowledge to the KB fostering faster learning of new tasks.
>
>     - The idea that tasks share common structure is a fundamental notion not only for our method, but for any method that expects to reuse or share information among tasks. As we mention in Section 2: “*there is no intersection between the classes of different tasks, but they share a similar domain. This domain intersection is key to take advantage of common patterns among the tasks.*”. We agree that if this is not the case, continual learning  methods -including our own- will not perform adequately.
>
> 2. We thank the reviewer for their scrutiny in our ablations. Regarding your suggestions:
>     - R1. The closest to what you ask for is our “Feature” ablation on the appendix section A.7, where we simply add a classifier on top of F_t, which would mix F_t + C_t without M_t. Results of that ablation are worse than MARK’s.  It is worth mentioning that the Mask functions consist of a linear layer with a ReLU activation, which does not greatly improve the capabilities of the “Feature” ablation.
>     - R2. This is effectively a good way to test whether the KB is relevant or not, we thank the reviewer for the useful comment. Given your suggestion, we ran this ablation for CIFAR100 over 3 seeds.  The result is an accuracy of 37.86%. This is a reduction in accuracy of  approx. 40% with respect to the full MARK model. We will add this result to our paper.
>
> 3. We thank you for the input, we have added a pointer at section 3 to facilitate reading.
>
> 4. To avoid confusion we will add clarifying text on our setup. In particular, we will explain that we are not working on an online setting, but rather a setting where we may use data from the current task freely, but once we move on to a new task it is not available. Regarding the use of a task id on inference, the answer is yes, it is needed. We will add text clarifying this as well.
>
> 5. We ran experiments on AGEM with the same hyperparameters as  our method, that is, using the same number of epochs on the data and a model with a similar amount of parameters to ours.  Our results on AGEM are similar to those reported in the original paper. The results shown for previous methods are on equal terms with ours, either in number of epochs, number of parameters, type of model, and optimizer. We will expand section A.2 on the appendix to include information about the hyperparameters used for each method. All experiments are run for 3 seeds (except when results are taken directly from a paper). Meanwhile, this is the list of hyperparameters we used for comparison of the baselines mentioned in the paper, plus the new ones asked by the reviewers:
>     - MARK: We use no scheduler, SGD, learning rate: 0.01, Momentum: 0.9, Weight Decay: 0.01. Batch size: 128. These settings are the same as for ACL. For the KB querying step: we train for 50 epochs, twice. For the KB update step: Inner Loop: 40 epochs, Outer Loop: 15 epochs. We set momentum to 0 in this step.
>     - ACL: we take results directly from the paper. They train for 200 epochs. Learning rate: 0.01, Batch Size: 128, SGD.
>     - HAT: we take results directly from the ACL paper. We assume hyperparameter choice between HAT and ACL to be fair as results are taken from the same paper. Therefore, it should be fair with MARK.
>     - AGEM:  For CIFAR100, we use an implementation from Avalanche [1]. We use a memory of 200 per task and run it for 100 epochs. For MiniImageNet we use the results from the ACL paper. This paper mentions that their AGEM implementation was modified to be multi-epoch. They used 200 epochs and 13 images per class for the memory. Learning rate: 0.01, SGD, Batch Size: 128.
>     - GPM: We slightly modified their code to run CIFAR100 with 20 tasks. We ran it for 100 epochs using their own code. Learning rate: 0.01, Batch Size: 128, SGD. Our KB model is very similar to theirs so we don’t replace it. We obtained Acc: 76.67% and BWT: -0.42%. For MiniImageNet, we take results directly from the paper. We were unable to find hyperparameters used either on the paper or code base.
>     - Experience Replay: We run it for 100 epochs. We use a memory of 250 elements per task. We use 250 to make sure that the memory requirements between ER and MARK are similar. Batch Size: 128, SGD, Learning Rate: 0.01. CIFAR100: 65.9% and -16.9% BWT
>     - SupSup: We used their code. Then, we tried different values of sparsity until we were able to replicate the same results of their paper with their setup. Using that sparsity value, we replaced their model with a model similar to our KB but with extra parameters, similar to the model used in ACL and GPM, and then we ran it for 100 epochs for 3 different seeds. Results are slightly below MARK: CIFAR100:  77.58% (1.3% std) and 0 BWT.
>
> 6. This is a very good question. On the one hand, as shown in previous work, an episodic metalearning scheme generates weights that generalize well and can be useful across tasks. On the other hand, as we mentioned above, the use of the mask selectors focus the gradient updates in the sources of new useful knowledge preserving previous one.
>
> [1] Lomonaco, Vincenzo, et al. "Avalanche: an End-to-End Library for Continual Learning." Proceedings of the IEEE/CVF Conference on Computer Vision and Pattern Recognition. 2021.

---

### Official Review · Reviewer_KoD1 · 2021-07-17

**Rating:** 5
**Confidence:** 4

**Summary:**

This paper is introducing a meta-learning framework into continual learning to promote learning resuable kwowledge from a sequence of tasks. The resuable kwowledge is simply defined by a set of shared model weights. The authors claim these weights as a common knowledege base (KB), which is not only reused but also incrrementally / continually enriched with new knowledge from new tasks, and propose to selectively / adaptively use them by utilzing some masking mechanism.

**Limitations And Societal Impact:**

- For the results, it would be good to compare the results with some possible upper bounds such as results from multi-task learning setting.
- for line 142, the naming is same as line 136, so it would be good to have another name for it.


**Main Review:**

- The sharable weights / modules are defined as a knowledge base (KB), but it doesn't really have some properties that the exact KB has, and this could probably be misleading naming.
- The role of KB is proposed to incrementally store meaningful knowledge into a set of weights and not only aovid catastrophic forgetting but also improve the learning of upcoming new tasks. To make it happend, the KB is updated through meta-learning based on a gradient-based method. (There were several works that similarly utilize a meta-learning framework to improve continual learning.)
- The ablation study shows that meta-learning reduces the forgetting problem. However, it is unclear why this meta-learning promotes to incrementally update the KB without catastrophic forgetting. The meta-learing objective is defined by mini-tasks that are sampled only from the current task, not caring about previous tasks.
- Moreover, the KB seems not to improve the learning of new tasks (i.e. accuracy), but act as a regularizer to avoid catastrophic forgetting. Because, the ablation study shows that the masking mechanism, which is highly related to feature modulation methods (i.e FiLM, DLN, CBN, SENet), is higly improving the accruacy.
- In terms of network architecutre, $F^t$, $M^t$ and $C^t$ are already a single classifier for task $t$. For example, if we asssume KB is very small or almost constant regardless of input $x$, then the mask variables become a representation itself. That way, it seems like the method is actually incrementally increasing the network size with having task labels.

**Time Spent Reviewing:**

6 hours

---

> ### Author Response · Authors · 2021-08-08
> **R1**
>
> We appreciate the comments received, below we answer each of them:
>
> Main Review:
>
> 1. We use the term KB to express more clearly the main intuitions behind the construction of our model. To avoid confusion, in the introduction of our work, we are explicit about the particular use of the term:  “*In this sense, the KB behind MARK is not given by an external memory that encodes information in its vectors, but by a trainable model that encodes shared information in its weights*”.
>
> 2. We will expand our current discussion of  OML and ANML (refs [43] and [44]) in our discussion of related work, explaining the main differences with respect to our work.
>
> 3. We understand the concern of the reviewer, the metalearning step (KB-update) is applied only to data from the current task, therefore, it is surprising its relevant effect to avoid CF, as shown in Figure 5. We believe that this is due to two main reasons: i) As shown in previous works, the use of an episodic training strategy fosters weight reusability. In other words, training based on an inner and outer loops that operate over mini-tasks, biases the learner to improve its ability to generalize to further tasks, fostering the learning of more generic rather than task-specific features. This impacts not only future but also previous tasks; ii) During metalearning, the feature selector masks are kept fixed, therefore, only the KB features that are relevant to solve the current task are updated. We believe that this is a key mechanism to mitigate CF because gradient updates are performed only where they really matter, helping to preserve previous knowledge. If an activation map is irrelevant for a certain task, then the value of the corresponding mask will be zero, making the gradient update associated to that activation being zero as well.
>
> 4. We agree with the reviewer, our results indicate that the metalearning step plays only a marginal effect to boost accuracy of the model in new tasks. However, as shown in Figure 4(a), it is worth mentioning that metalearning plays a relevant role to enrich the KB, in the sense that it helps to learn tasks faster and in a stabler fashion. This indicates a synergistic combination between the metalearning and mask training steps that leads to improve model accuracy.
>
> 5. - This is a relevant insight from the reviewer. Indeed, in Table 1, we include results when Ft is initialized with features from Resnet pre-trained on ImageNet. The use of these features leads to a large boost in performance, as expected, mainly in the Mini-Imagenet dataset. However, Table 1 also shows that in this case,  where the model prefers a solution that relies more on Ft than KB, the problem of CF increases as measured by the change in BTW score. We believe that this helps to support the relevance of a synergistic combination between the metalearning and mask training steps.
>    - With respect to model complexity, it is worth mentioning that in our experiments we use an equivalent amount of parameters to compare our model with respect to the baselines, as is shown in section A.4 of the Appendix.
>
> Limitations:
>
> 1. We have added this upper bound as part of the baselines. Results are: CIFAR100: 84.04% STD: 1.2% BWT: 0%
>
> 2. We use the same naming because both steps refer to the same procedure with no differences, however, we agree that it is confusing, therefore, we will clarify the issue and adjust the labels to facilitate the reading.

---

### Decision · Program_Chairs · 2021-09-28

**Decision:**

Accept (Poster)

**Comment:**

The paper proposes a continual learning method that maintains a knowledge base (KB) shared across all tasks and utilizes a form of meta learning to update the KB and dynamic feature masking to selectively choose relevant weights for each task from the KB. The three training stages are explained, and empirical results are provided that show competitive performance to alternatives. Additionally, an ablation study is provided showing that meta learning is most important for reducing catastrophic forgetting while the masking mechanism contributes to achieving higher accuracy.

The proposed method is an interesting combination of techniques that end up working synergistically to improve both overall performance and mitigate catastrophic forgetting. While the ablation study helps give an idea about which component more significantly contributes to what performance aspect, the writeup may benefit from more explanation  of the design choices and the intuition we should gain from the ablation study, making it more clear why the method works. The experiments section could be improved by clarifying the evaluation setup and comparing all methods on the same setup. Some of these aspects were discussed in the rebuttal, convincing some reviewers to update their scores.


**Consistency Experiment:**

NeurIPS has a long history of experimentation. In 2014, NeurIPS ran an experiment in which 10% of submissions were reviewed by two independent committees to quantify the randomness in the review process. This year, we repeated a variant of this experiment to see how the quality of the review process has changed over time.  This paper was part of the experiment and was therefore assigned to two committees (consisting of reviewers, an Area Chair, and a Senior Area Chair) that reached independent decisions.  If both committees made the same recommendation, this recommendation was followed. If a single committee recommended acceptance, the paper was accepted (with the exception of a few cases in which the other committee identified what we considered a fatal flaw, e.g., an error in a key result).

This copy’s committee reached the following decision: **Accept (Poster)**

The other committee assigned to the paper recommended **Reject**.  You can find the other set of reviews, along with any follow up discussion with the authors here:
https://openreview.net/forum?id=sFyrGPCKQJC